# Regional-Scale Forest Mapping over Fragmented Landscapes Using Global Forest Products and Landsat Time Series Classification

**Viktor Myroniuk** [1], **Mykola Kutia** [2,*], **Arbi J. Sarkissian** [2], **Andrii Bilous** [1] **and Shuguang Liu** [3]

1   Department of Forest Mensuration and Forest Management, National University of Life and Environmental Sciences of Ukraine, Heroiv Oborony Str. 15, 03041 Kyiv, Ukraine; victor.myroniuk@nubip.edu.ua (V.M.); bilous@nubip.edu.ua (A.B.)
2   Bangor College China, Joint Unit of Bangor University, Bangor, UK and Central South University of Forestry and Technology, 498 Shaoshan Rd., Changsha 410004, Hunan, China; a.sarkissian@bangor.ac.uk
3   College of Life Sciences and Technology, National Engineering Laboratory of Applied Technology for Forestry and Ecology in South China, Central South University of Forestry and Technology, 498 Shaoshan Rd., Changsha 410004, Hunan, China; shuguang.liu@yahoo.com
*   Correspondence: m.kutia@bangor.ac.uk; Tel.: +38-097-70-60-093

**Abstract:** Satellite imagery of 25–30 m spatial resolution has been recognized as an effective tool for monitoring the spatial and temporal dynamics of forest cover at different scales. However, the precise mapping of forest cover over fragmented landscapes is complicated and requires special consideration. We have evaluated the performance of four global forest products of 25–30 m spatial resolution within three flatland subregions of Ukraine that have different forest cover patterns. We have explored the relationship between tree cover extracted from the global forest change (GFC) and relative stocking density of forest stands and justified the use of a 40% tree cover threshold for mapping forest in flatland Ukraine. In contrast, the canopy cover threshold for the analogous product Landsat tree cover continuous fields (LTCCF) is found to be 25%. Analysis of the global forest products, including discrete forest masks Global PALSAR-2/PALSAR Forest/Non-Forest Map (JAXA FNF) and GlobeLand30, has revealed a major misclassification of forested areas under severe fragmentation patterns of landscapes. The study also examined the effectiveness of forest mapping over fragmented landscapes using dense time series of Landsat images. We collected 1548 scenes of Landsat 8 Operational Land Imager (OLI) for the period 2014–2016 and composited them into cloudless mosaics for the following four seasons: yearly, summer, autumn, and April–October. The classification of images was performed in Google Earth Engine (GEE) Application Programming Interface (API) using random forest (RF) classifier. As a result, 30 m spatial resolution forest mask for flatland of Ukraine was created. The user's and producer's accuracy were estimated to be 0.910 ± 0.015 and 0.880 ± 0.018, respectively. The total forest area for the flatland Ukraine is 9440.5 ± 239.4 thousand hectares, which is 3% higher than official data. In general, we conclude that the Landsat-derived forest mask performs well over fragmented landscapes if forest cover of the territory is higher than 10–15%.

**Keywords:** flatland Ukraine; forest cover map; forest mask; fragmented landscapes; global forest change; Google Earth Engine; Landsat imagery; random forest algorithm

## 1. Introduction

Reliable data on forest cover addresses a wide range of issues, particularly in informing management decisions related to land-use and land-cover change. In recent decades, satellite imagery

collected by the Landsat platforms in forestry applications has increased owing to the availability of long time series of satellite observations of land cover and its dynamics. This has helped in developing global forest cover products and advancements in automated methods for thematic image processing. Changes in access policies to Landsat imagery archives [1] have resulted in several fine spatial resolution forest cover products (e.g., forest masks) at global scales [2–4]. Thus, dense time series of Landsat images have been recognized as an effective tool for monitoring the dynamics of forested landscapes [5]. These time series images enable us to estimate forest conditions caused by intra-annual variation in phenology and facilitate improved classification. Previous studies have proven the superiority of satellite-based methods for characterizing and mapping forest vegetation under different climatic conditions. Continuous improvements made in remote sensing technologies have motivated researchers to summarize recent advances in application time series, ranging from land cover classification [6] and tree species composition classification [7], to predicting detailed forest inventory parameters [8].

Landsat images (Thematic Mapper (TM), Enhanced Thematic Mapper Plus (ETM+), Operational Land Imager (OLI)) are regarded as the standard Earth observation data for ecological monitoring over large scales, and provide unique opportunities for assessing changes in forest cover [9,10]. Landsat imagery is crucial for revealing trends in land cover changes and forest disturbances [11,12]. The significant interest of forest resources assessment using Landsat images is explained by the fact that forests could be mapped with higher accuracy compared to other types of vegetation. An important role of the Landsat platform is its development of methods for the global monitoring of forest ecosystems [13]. As a result of national programs and international initiatives, several remote sensing-based products have been released that characterize forest cover at regional [14], continental [15], and global [2–4] scales. For example, the JAXA forest/non-forest (JAXA FNF) mask was the first high spatial resolution (25 m) global forest map developed using synthetic aperture radar (SAR) data [16].

Proportional per-pixel estimates of tree cover represent important improvements of thematic representation of forests over discrete maps. In 2013, two fine scale global maps were released, characterizing forests in a continuous form [3,4]. Both global forest change (GFC) and Landsat tree cover continuous fields (LTCCF) predict tree cover while forest cover should be considered for accurate mapping, and to avoid misclassification in some landscapes [17]. Compared to its predecessors, i.e., coarse-scale Advanced Very-High-Resolution Radiometer (AVHRR) and Moderate Resolution Imaging Spectroradiometer (MODIS) vegetation continuous fields, Landsat-based percent tree cover global products reveal 100–1000 times more detailed information. In particular, the spatial resolution of 30 m is essential for forest dynamics assessment because many forest cover changes are detected at this scale. Assessing the accuracy of Landsat-based tree cover products has been the focus of recent research [18]. Methods for the evaluation and usage of the GFC map [3] have been discussed in terms of reporting forest cover estimates at the national level [19,20]. Recommendations for the design and implementation of a sampling frame using GFC data for map area assessment were provided as a practical guide [21].

Repeated observations have decreased the probability of occurrence-missing observations and advanced remote sensing techniques in several ways. Change metrics derived from time series allows for depicting seasonal differences between forests and other types of vegetation, yielding an improved classification of tree species composition [22,23]; predicting forest productivity [24]; and determining forest structure and aboveground biomass [25,26]. There are good examples in the literature of using time-series satellite images for mapping species composition [27,28]. Chrysalis et al. [29] have indicated the importance of image acquisition dates for the mapping and estimation of forest parameters. The authors concluded that multitemporal classification algorithms overperformed single-date as well as one-seasonal approaches. Other studies have also shown that methods of image processing capturing annual leaf phenology cycles contribute more to the accuracy of classification than the spatial resolution of remote sensing data [22,23,30]. Employing data-rich time series approaches has therefore become

an important development in this field and is predicted to replace bi-temporal image comparisons relatively soon [13].

The large and continuously updated capacity of Landsat data facilitates the development of MODIS-like methods for image composition. These are based on the analysis of time series aiming to select the best available observations for each pixel over a study period [31]. The composing approach could preferentially select the "greenest" dates having the maximum normalized difference vegetation index (NDVI) value [32] or median pixel value at each spectral band [33]. Maximum value composites have been proven to overcome the cloud shadowing problem but are likely to be skewed towards the greenest dates. For the purposes of detecting trends in forest cover disturbances, the medoid composing approach [34] was proposed by some authors [35]. In terms of mapping forests using dense time series of Earth observations, different principles for extracting spectral metrics were examined. Hansen et. al [32] demonstrated an approach in which spectral inputs for classification were extracted using maximum NDVI composites for annual, summer, and autumn temporal mosaics, but specific statistical rules (minimum, maximum, median values, 1st and 3rd quartiles) were employed to capture seasonal phenology for growing season mosaic. Parente and Ferreira [36] have recently found that spectro-temporal metrics derived from the percentile analysis of yearly NDVI values are associated with seasonal responses of vegetation. We agree with the authors that this greatly enhances the ability to detect dynamics in forest cover through the season and to produce more accurate tree species composition maps.

The scale and detail of mapping have gradually evolved with development of finer spatial and spectral resolutions of sensors. Freely available imagery such as Landsat, Sentinel, and commercial Earth observation data (PlanetScope, RapidEye, SPOT etc.) have been applied for mapping large areas of diverse land cover types [6,37]. Remote sensing technologies have been evolving rapidly, including medium to very high resolution (VHR) optical sensors, SAR, and LiDAR (or Light Identification, Detection and Ranging), which allows for more accurate land-cover mapping [8]. With the advancement of this technology, multi-source remote sensing data fusion techniques are emerging that require new classification algorithms [38]. When dealing with normally distributed unimodal data, supervised parametric classification algorithms, e.g., maximum likelihood classification (MLC), deliver robust results. However, they are not suitable when processing multi-modal input datasets due to underlying assumptions of the normal distribution of data [39]. In contrast, non-parametric supervised classifiers do not make any assumptions regarding frequency distribution and have therefore become increasingly popular for classifying remotely sensed data, which rarely have normal distributions. Given this technical advantage, popular non-parametric classifiers such as the classification and regression tree (CART), random forest (RF), support vector machine (SVM) [40], and artificial neural network (ANN) have recently been adopted for land-cover mapping [41].

Numerous spectral features are used to map forests using time series. Banskota et al. [5] have provided at least four major groups of such features, namely: spectral band data, spectral ratio indexes, tasseled cap transformation (TCT) bands, and spectral mixture indexes. Among modern non-parametric modelling algorithms, RF has been used extensively in forest mapping practices due to its robustness and ability to process a large number of predictor variables [42,43]. Previous studies on forest vegetation mapping have indicated the need for reducing input-variable lists based on the estimation of variable importance in RF regression algorithms [29]. Other studies highlight the utility of transforming original spectral bands into the three TCT components of brightness, greenness, and wetness [33]. However, nonparametric approaches for image classification perform well even if input datasets include a significant number of predictors [15,26].

Dense time series demonstrates a high capacity for mapping forests but needs significant investment in data processing. The calibration of classification algorithms requires extensive computing resources, thus presenting a limiting factor for many users. The advent of the Google Earth Engine (GEE) cloud-based computing [44] in recent years has helped facilitate broad-scale studies using high-performance mapping algorithms [36,45,46]. GEE enables easy access to different

publicly-available datasets, including the collection of Landsat pre-processed images, and dramatically reduces the time required for generating accurate maps. GEE has also become highly effective for classifying multi-temporal satellite imagery, global geospatial analysis, and visualization [33,47].

Recently, there has been an increasing need to understand better the importance of landscape fragmentation processes [48,49]. Forest fragmentation refers to changes in forest cover and can be measured using land-cover raster maps derived from satellite imagery. Urbanization, the conversion of forested lands into agricultural lands, transportation infrastructure expansions, intensive forest management are accepted as the major human activities causing fragmented landscapes [50]. In the flatlands of Ukraine, fragmented forests induce potential errors during the process of forest cover mapping using classification algorithms. Furthermore, assessing the extent of fragmented forests can help to inform policy and decision making for forest management practices [51]. Thus, forest mapping over fragmented landscapes is essential at the regional-scale.

The literature highlights some applications of remotely-sensed datasets and dense time series of optical satellite images for global and nation-wide forest mapping. However, the application of these datasets for characterizing a regional-scale pattern of forest cover has been limited. The aims of this study were: (1) to examine the consistency of site-specific accuracy between four global forest products at 25–30 m spatial resolution in fragmented landscapes in the flatlands of Ukraine and define their utility for reporting forest area estimates, and (2) to investigate the role of dense Landsat 8 OLI time series for producing a regional-scale forest mask. We also tested the performance of the cloud-based GEE platform for seamlessly processing large amounts of spectral and thematic data with potential to apply the platform for larger scales in future. This is perhaps the first attempt at comparing the performance of global Landsat-based percent tree cover maps, discrete forest masks, and classification time series from OLI observations over severely fragmented landscapes in Ukrainian flatlands.

## 2. Materials and Methods

### 2.1. Study Area

The study area covered 541,000 km$^2$ (or approx. 90%) of Ukraine in Eastern Europe. This area encompasses nearly all flatland landscapes of the country. The elevation ranges between 0–400 m.a.s.l (from the southernmost point of the Black Sea lowlands to the Pre-Carpathian region in the west). Land use is highly heterogeneous with multiple cropping systems that have historically defined the distribution of forests. Forest cover varies substantially along a latitudinal gradient from 20–35% in northern Polissya to the central forest-steppe zone, down to 4–6% in the steppe zone in the south (Figure 1). The forests are impacted by different natural and anthropogenic factors that predefined the mosaic pattern and combination of woody and open landscapes. The forests in Ukrainian Polissya are fragmented due to clearcut harvesting and salvage logging [52]. Only narrow linear windbreaks and preserved small patches of natural forests in river valleys have remained in the steppe zone. Coniferous and hardwood deciduous stands occupy nearly 85% of the forested area in the region. More than 90% of coniferous forests are composed by Scots pine (*Pinus sylvestris* L.). Common oak (*Quercus robur* L.) dominates among deciduous hardwood species while ash (*Fraxinus excelsior* L.) and hornbeam (*Carpinus betulus* L.) together occupy less than 5% of the total area. Silver birch (*Betula pendula* L.) and alder (*Alnus glutinosa* (L.) Gaerth.) are major softwood deciduous tree species that, together with some other rare species, cover the remaining 15% of forested area.

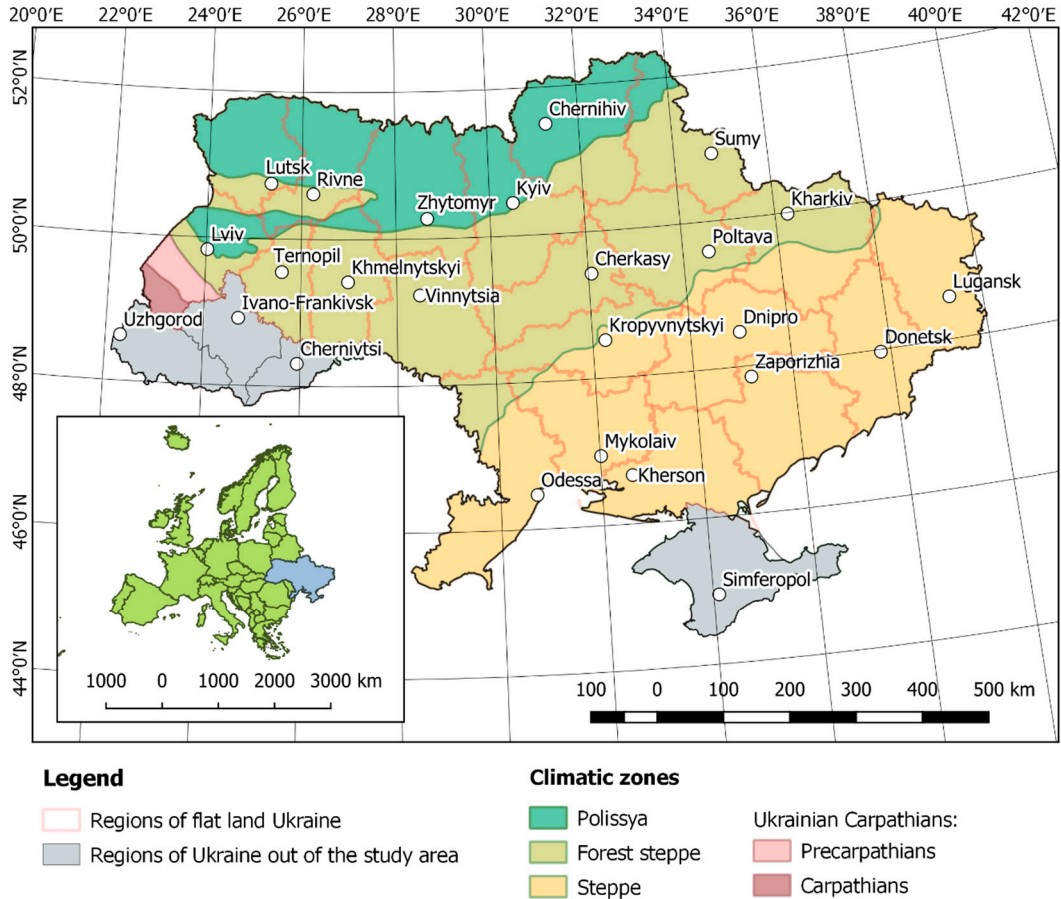

**Figure 1.** The study area representing three main climatic zones of flatland Ukraine.

## 2.2. Remote Sensing Data

### 2.2.1. Global Forest Products

We analyzed four global forest products of 25–30 m spatial resolution: (1) global forest change [3], (2) Landsat Tree Cover Continuous Fields [4], (3) JAXA forest/non-forest mask [16], and (4) GlobeLand30 [2]. The GEE Application Programming Interface (API) provided direct access to GFC and LTCCF while the other two datasets were uploaded as user assets into the platform.

Global forest change (GFC) is an open-access global map of forest cover changes developed at the University of Maryland [3]. This product is based on the radiometrically corrected Landsat 5 TM and Landsat 7 ETM+ images for the year 2000, which were cleared from clouds, shadows, and water bodies. Image processing was performed in the form of per-pixel composite mosaics, which allowed to derive a series of phenological metrics for the classification. Forest vegetation was defined in each 30 × 30 m pixel as trees taller than 5 m, and mapped in the "treecover 2000" layer of the GFS dataset as per-pixel estimates of percent canopy cover. The GFC dataset includes additional layers that reflect total ("loss") and annual ("lossyear") forest loss, and reforestation ("gain") for the period from 2000.

Landsat tree cover continuous fields (LTCCF) is a global product of continuous estimates of percentage of woody vegetation cover (taller than 5 m) produced at the University of Maryland [4]. The LTCCF was developed for the years circa- 2000, 2005, 2010 using a tree-cover rescaling algorithm. The per-pixel canopy cover was predicted at 30 × 30 m using Landsat 5 TM and Landsat 7 ETM+ radiometrically corrected, cleared from clouds and shadows reflectance data and the vegetation cover estimates derived from MODIS vegetation continuous fields as a reference. Thus, the spatial resolution of each Landsat scene was aggregated to a spatial resolution of 250 m to train the classifier, followed by the fitted model applied to the original 30-m Landsat data.

Global PALSAR-2/PALSAR Forest/Non-Forest Map (JAXA FNF) is a global forest/non-forest mask generated by Japan Aerospace Exploration Aerospace Agency [16]. The JAXA FNF is the first 25-m spatial resolution global forest mask that was developed using active remote sensing, specifically ALOS L-band SAR HH and HV polarization data. The forests were classified by applying regional-specific backscatter thresholds to produce annual global mosaics for 2007–2010, the global map of forest and non-forest cover, as well as maps of forest loss and gain.

GlobeLand30 is a global map of the land cover developed in the National Geomatic Center of China [2]. The map was created using radiometrically and geometrically corrected Landsat TM and ETM+ satellite images with a spatial resolution of 30 m. For the classification, automated approaches of per-pixel processing augmented by object-based methods with expert knowledge of the natural (form, geographical location, location) and cultural (types of land use) parameters of different types of global land cover was developed. The classification introduced ten thematic land cover classes including forest cover for the years 2000 and 2010.

### 2.2.2. Spectral Data

The remote sensing dataset consisted of 1548 Landsat 8 OLI images selected over the study area for the period 2014–2016. Due to the large volume of information, GEE API was used for nearly all phases of data processing. We focused on the Landsat Level-1T Top of Atmosphere (TOA) collection and combined images into four seasonal mosaics: yearly, summer, autumn, and April–October. To improve the quality of spectral data, we expanded the timeframe for image acquisition as well as filtered-only images having cloud coverage of less than 20%. As a result, for each WRS-2 path/row frame, 20–30 scenes of Landsat images were selected. The contribution of satellite images acquired during 2014 and 2015 was approximately equal, but only 465 scenes were selected for 2016.

Atmospheric conditions have significant impacts on the quality of Landsat data. The most robust approach for the radiometric correction of Landsat imagery is the LEDAPS (the Landsat ecosystem disturbance adaptive processing system) [53], which estimates per-pixel surface reflectance. Wulder et al. [54] reported that the surface reflectance imagery is vital in studies aimed at the detection of changes in land cover. However, TOA correction was considered for a long time as a critical step towards the inter-scene normalization of Landsat images and has been successfully applied in land cover mapping [13,15,32,33]. In the current study, we focused on TOA reflectance images because the Landsat surface reflectance products had only just become available through GEE in 2016, when the major phases of the analyses were performed.

In the GEE environment, we applied a built-in cloud scoring algorithm to the selected Landsat images for cloud screening. This assumes that clouds have higher reflectance in blue and all visible as well as near-infrared bands, but reasonably lower reflectance in thermal bands. The algorithm uses the normalized difference snow index (NDSI) to distinguish clouds and snow cover. We masked clouds if cloud scores exceeded an 80% threshold. Images selected for the abovementioned seasons were composed in seamless mosaics using a maximum value composite (MVC) approach. We followed experiences published in previous studies [15,32] and combined images selecting only those pixels that had maximum NDVI value for composing period. As a result, this allowed us to preferentially select the greenest pixels and exclude cloud shadows from composites. However, only yearly, summer and autumn mosaics were composed by this approach (Figure 2). The April–October mosaic was incorporated in the study because it coincides with leaf-on periods in Ukraine. To address the phenology phases in image classification, we selected pixels by applying some statistical rules: minimum and maximum values, 1st and 3rd quartiles, and median of TOA reflectance. The feature space for classification was constructed using TOA reflectance from visible, near-(NIR), shortwave-(SWIR) and thermal-infrared (TIR) bands: Band 4 (0.64–0.67 μm), Band 5 (0.85–0.88 μm), Band 6 (1.57–1.65 μm), Band 7 (2.11–2.29 μm), Band 10 (10.60–11.19 μm), Band 11 (11.50–12.51 μm). For Landsat TOA images, we also calculated simple band ratios as follows: Band 4/Band 5, Band 4/Band 7, Band 5/Band 7. Finally, the NDVI index was determined. Three bands (brightness, greenness, wetness) of tasseled cap

transformation (TCT) were derived using the coefficient empirically derived for the Landsat 8 OLI sensor [55].

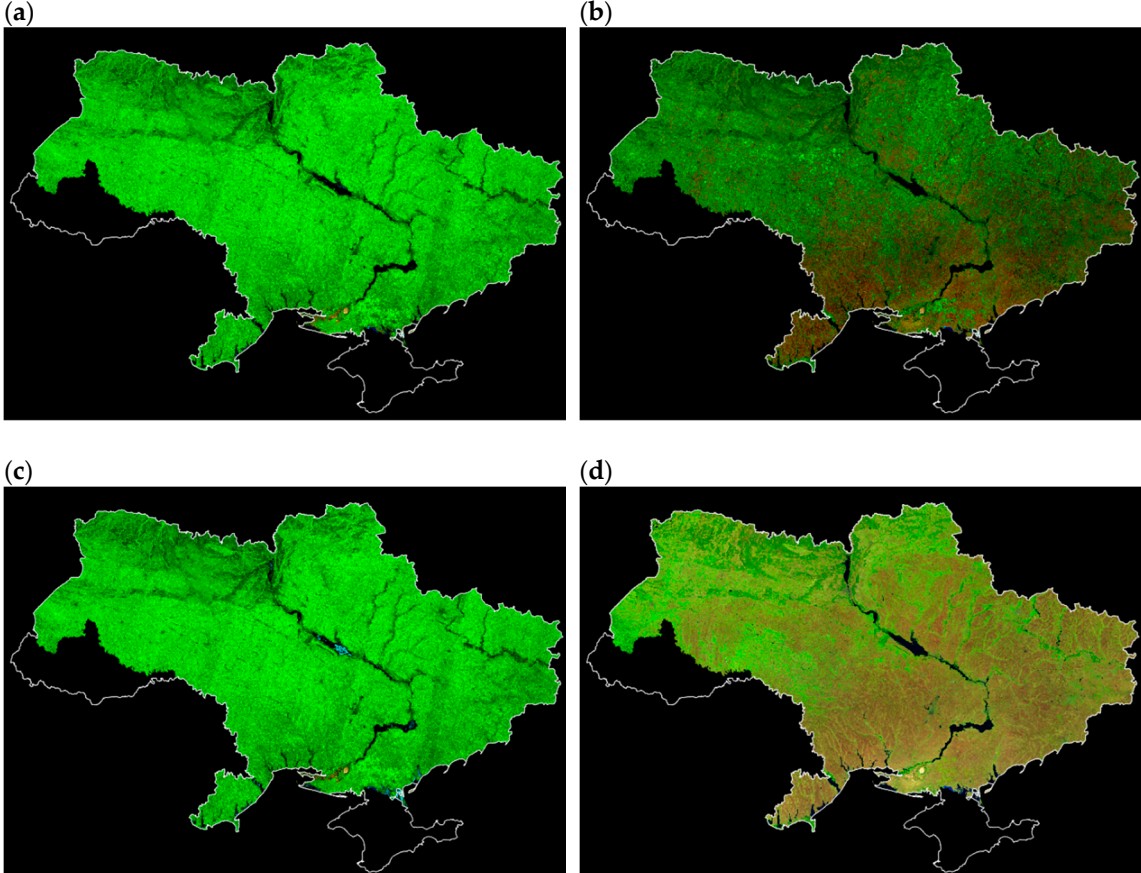

**Figure 2.** RGB images of seasonal composited mosaics (band combinations 6–5–4) of Landsat 8 OLI images for the flatland Ukraine and country outline (white line): (**a**) summer; (**b**) autumn; (**c**) yearly; (**d**) April–October. MVC composites were used for the visualization of (**a**), (**b**), and (**c**), and 1st quartiles of pixel values derived from time series applied for (**d**).

In this study, we focused on Landsat-derived spectral features and indices that are commonly used for characterization of vegetation phenology and forest dynamics [5]. Thus, the SWIR band is widely applicable because of its sensitivity to vegetation moisture content and canopy density. NDVI is a ratio of NIR and red-band wavelengths, which correlate with many biophysical parameters of forests, such as live biomass. Additionally, NIR and TIR bands are highly affected by snow cover. Moreover, the TCT bands are associated with physical parameters of land cover and widely applied in ecological monitoring studies [27].

Dense time series of Landsat data addresses phenology-based methods for image classification that depict the variation of spectral features in the yearly timeframe (Figure 2). The mosaics mark seasonal patterns of land use and track phenological events, i.e., crop conditions, vegetation growth, senescence, and dormancy, thus enhancing the spectral difference between vegetation classes. The trends of high, medium, and low pixels' values in infrared bands is depicted by percentiles derived from time series for the period of April–October.

### 2.3. Reference Data

In our study, we developed two independent reference datasets using stratified random sampling based on the guidelines for designing a sampling frame provided by Olofsson et al. [21,56]. Following these recommendations, we adopted a stratified random sampling protocol that is very practical in

terms of adequate reporting on overall map accuracy as well as class-specific accuracy estimates. This design is advantageous in remote sensing in cases where the strata are of interest for reporting. We employed a proportional-to-class allocation so that at least 50 sample units would fall into classes that occupy a small area in the region. Thus, we slightly shifted the distribution from the proportional allocation by increasing the sample size for rare classes. This allowed us to reduce standard errors in accuracy estimation for "forest" class in a number of southern and eastern regions in flatland Ukraine with low forest cover. We calculated the sample size distribution within mapped classes based on formula #13 in Olofsson et al. [21] to reach the 2% standard error, and taking into consideration the mapped proportions of each class and standard deviation for each stratum (as estimated using assumed user's accuracy of 0.9 for large classes and 0.7 for rare classes).

The first dataset was used for accuracy assessment of the global forest masks. For stratification, we employed the GFC map that had information on forest change since 2000. Applying a 25% threshold for canopy cover where forest loss and gain had not occurred, we created two major stable classes: (1) permanent forest and (2) permanent not-forest. Since forest cover is highly influenced by anthropogenic and natural disturbances, it was also important to get enough information for land cover classes where forest change took place. Thus, we created two small change classes characterizing deforestation and forest regeneration. They were both extracted from corresponding loss and gain layers of the GFC dataset. The sample size for each of the 21 administrative regions (i.e., oblasts) of flatland Ukraine varied from 220 to 250, assuming the standard error of the overall accuracy would achieve less than 2% out of a yielded total of 4700 sample units.

All samples were visually interpreted using Google's satellite images with submeter spatial resolutions. For this task, we used the Collect Earth plugin feature for Google Earth that simplified image analysis and the collection of reference data [57]. To match the minimal mapping units adopted in forest inventory of Ukraine for the delineation of forest stands, we considered each sample unit as a 0.25 ha plot. In cases where plots simultaneously covered different land cover classes or forest stands, a central point class was assigned to these plots. We analyzed satellite images in time to better distinguish between coniferous and deciduous forests for the reporting period of 2015. The hierarchical image interpretation scheme was adopted that included eight first-order thematic classes (water bodies, wetlands, settlements, other unproductive lands, croplands, grasslands, shrublands, forests) subdivided into at least three second-order subclasses. For tree species composition, we divided forest stands into three groups based on a portion of grid points that were overlaid on sample unit: (1) coniferous, (2) deciduous (if the proportion of respectively coniferous or deciduous tree species in canopy cover was 75 % or more), and (3) mixed. As an additional attribute, we included the image date used for interpreting the sample points. This is a useful option available in Google Earth (Open Foris plugin) for the correct analysis of the GFC map (losses and gains) and classification of Landsat images, which allowed us to filter samples interpreted only for the 2015-time frame. Examples of certain aspects of image interpretation are provided in Supplementary Figure S1.

The first reference dataset was also used for the classification of Landsat imagery over flatland Ukraine. However, a new independent dataset was created for assessing the accuracy developed from the Landsat 8 OLI data land cover map. The procedure of reference data collection for accuracy assessment was the same as for assessing global products; however, the stratification was performed based on the eight land cover classes. We treated wetlands, shrublands, water bodies, settlements, and other unproductive areas as rare classes that occupy a relatively small area in comparison with croplands, grasslands, and forests. Given that we had more thematic classes that were treated as strata, the sample size of the second reference dataset for each region ranged from 410–460 sample points and totalled 9237 for 21 oblasts of flatland Ukraine.

A forest inventory database (FID) containing estimates of attributes for each forest stand was also employed in the study. We selected 15 forest enterprises equally distributed over flatland Ukraine to test a per-pixel accuracy of percent tree cover that GFC and LTCCF data provided.

## 2.4. Forest Mapping Approach

Forest cover on maps are more frequently represented as discrete classes. "Continuous fields" is an alternative characterization approach that refers to the fraction of pixel area occupied by forests. In terms of interpretation, continuous maps provide greater flexibility, allowing experts to select an adequate threshold for the disaggregation of forested and unforested areas [58]. The method of estimating forest area using continuous attributes (e.g., canopy cover) creates considerable opportunities in forest mapping. Thus, choosing a various threshold value of canopy cover helps in defining what is a "forest". For example, Sannier et al. [20] proposed selecting this value by controlling the accordance of forest area estimates on the test areas uniformly distributed throughout the country. The proposed technique is based on a comparison of precisely mapped forest area within a network of sampling units and its estimates are derived from global datasets using the national definition of forest. In the case of overestimation, the threshold would gradually increase, and in the case of underestimation the threshold would be lowered. Schepaschenko et al. [17] have calibrated a country-specific threshold for classifying hybrid percent tree cover map by matching the total forest area for each country with national statistics. One of the major drawbacks of the foresaid approaches for our study is that the national definition of forest in Ukraine relies on relative stocking density (0.3 and more), which could not be directly calculated from global forest maps. Thus, we have investigated the relationship between tree cover values at the pixel-level and actual relative stocking of forest stands from the forest inventory records to justify a threshold for the classification of continuous global forest maps GFC and LTCCF into binary maps. To derive the forested area from the JAXA FNF and GlobeLand30 maps, we applied a binary classification approach. Figure 3 depicts the datasets and major steps of the analysis used in our study.

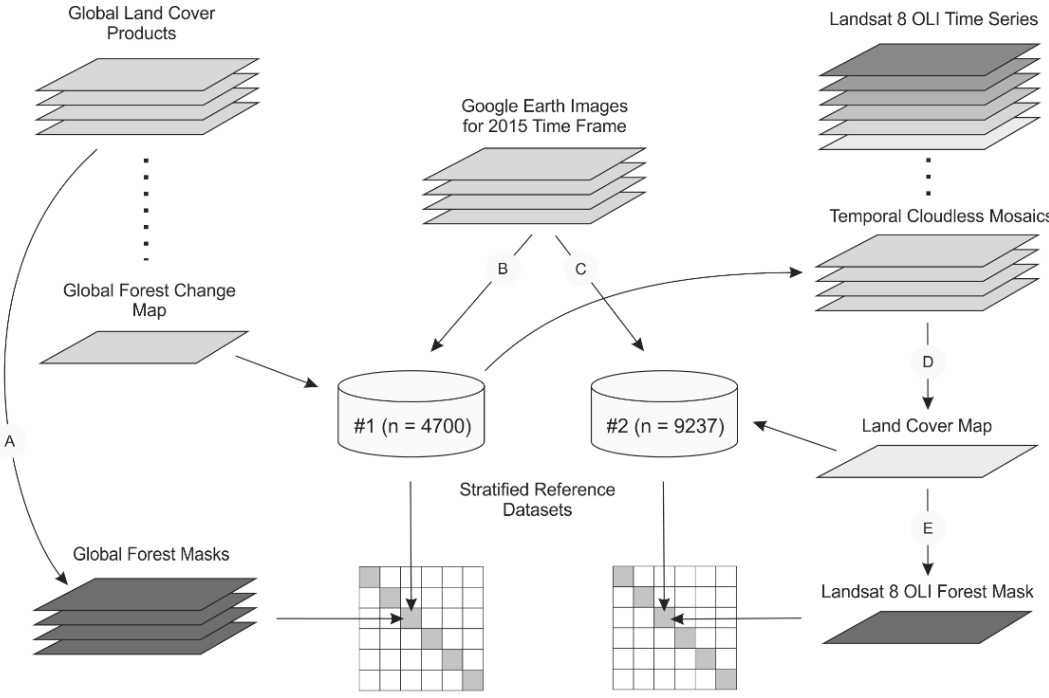

**Figure 3.** Flow-chart illustrating the use of global forest products and Landsat 8 OLI satellite imagery in the current study: A–threshold (GFC and LTCCF) or binary (JAXA-FNF and GlobeLand30) classifications; B,C–visual interpretation of historical Google Earth images for training and validation; D–random forest classification; E–binary classification.

Examining the relationship between canopy cover and relative stocking density of forest stands is necessary for the proper interpretation of the global percent tree map for Ukraine. In the present

study, we refer to the biophysical definition of forests as an area covered by woody vegetation having a predefined minimum canopy cover. According to Ukrainian regulation, the criteria for classifying forested and unforested areas is relative stocking density of 0.3 estimated as a ratio between the sum of the basal area of each plot to the basal area of the "normal stand". To perform these comparisons for the period of 2001–2015, we updated the canopy cover information derived from the GFC map by flagging all pixels where forest loss occurred as unforested and assigned a canopy cover of 25% for those pixels where there was a gain in forests. This was the same threshold value we used for the forest mapping based on the LTCCF product.

The GFC provides data on forest dynamics starting from the year 2000. As the criteria for defining changes i.e., transformation of forested to unforested areas and vice versa, vegetation taller than 5 m in height and 25% tree canopy thresholds were chosen [3]. Due to the specifics of the distribution of forested areas by tree species composition and its proportion in different climatic zones of the study area, we analyzed the performance of the GFC map by applying thresholds for classifying tree cover in order to identify the most suitable value for the study area. To find the specific threshold value of canopy cover for the study region that depicts the difference between forested and unforested lands, we extracted 10,000 plots (forest stands) from the forest inventory database having a relative stocking density of 0.3–0.5. Then, we mapped them over the updated 2015 GFC datasets to get the canopy cover values.

## 2.5. Random Forest Classification

Due to the large volume of spectral information and potential predictor variables, only nonparametric methods for image classification were considered in the case of processing seasonal Landsat mosaics. We chose the RF classifier which has been used extensively in Landsat time series studies [29,33]. The RF method [59] belongs to the group of ensemble learning algorithms in which the best results from multiple classification trees are selected using the votes. Training of the RF classifier was performed using the bootstrap aggregating procedure (or bagging) in which about two thirds of sample size and the other one third of observations were used for calculating the unbiased assessment of out-of-bag (OOB) error. In the present study, we applied the randomForest package [60] for statistical R software [61] for analysis. One of the advantages of the RF classifier is that it calculates the variables' importance using the out-of-bag (OOB) sample. As a result, the percentage increase in classification error (%IncMSE) will show if the corresponding variables were to be excluded from the training dataset. Many studies have shown how RF could be applied for optimizing a set of predictor variables [29,62,63].

We composed over 50 spectral band and band ratios of Landsat 8 OLI time series in four seamless mosaics (Table 1). In order to reduce the dimension of the feature space, we visually inspected all bands and found that minimum pixel values in the April–October mosaic refer to shadowed areas, while maximum values show artifacts that remained after cloud filtering. Afterwards, we used the RF classifier and %IncMSE as a variable importance measure and ranked variables by their values.

**Table 1.** The list of predictor variables extracted from seasonal mosaics of Landsat 8 OLI images.

| Spectral Features Selected for Yearly, Summer, Autumn TOA Reflectance Seasonal Mosaics | Spectral Metrics Derived for April-October TOA Reflectance Seasonal Mosaic |
|---|---|
| Band 4, Band 5, Band 6, Band 7, Band 10 NDVI | Minimum and maximum values for: Band 4, Band 5, Band 6, Band 7 and NDVI |
| Band 4/Band 5 ratio Band 4/Band 7 ratio | 1st and 3rd quartiles for: Band 4, Band 5, Band 6, Band 7 and NDVI |
| Band 5/Band 7 ratio TCT: Brightness, Greenness, Wetness | Median values for: Band 4, Band 5, Band 6, Band 7 and NDVI |

Previous research [62] has shown that increasing the number of classification trees (*ntree*) improves the stability of RF predictions, but incorporating more variables for splitting trees at each node (*mtry*) enhances the discrimination between important and irrelevant variables. Thus, we used *ntree* = 500 as recommended by Belgiu and Drăguţ [64] and many other studies [62,65], but the optimal *mtry* values of the RF classifier were estimated using the tuneRF function available in the randomForest package for R. The calculation was performed separately for each seasonal mosaic and all mosaics combined. As an option, classification with and without geographical coordinates was considered. We have included geographic location variables to test their performance as an indirect measure of environmental factors that influence vegetation distribution over flatland Ukraine.

The OOB error was estimated as a mean value from 50 runs of the RF algorithm and is presented in Table 2. The higher accuracy of classification was obtained for the combined mosaic that includes data from four selected seasons augmented by longitude and latitude (OOB = 25.4%) followed by combined mosaic without geographical coordinates (OOB = 25.6%).

**Table 2.** Accuracy assessment analysis of land cover classification using different seasonal mosaics.

| Seasonal Mosaic | *mtry* | Spectral Data | | Spectral Data and Geographical Coordinates | |
|---|---|---|---|---|---|
| | | Number of Predictors | OOB Error, % | Number of Predictors | OOB Error, % |
| Yearly | 6 | 12 | 36.5 | 14 | 33.8 |
| April-October | 8 | 15 | 26.7 | 17 | 25.7 |
| Summer | 6 | 12 | 36.8 | 14 | 34.6 |
| Autumn | 6 | 12 | 33.0 | 14 | 31.6 |
| Combined | 14 | 51 | 25.6 | 53 | 25.4 |

To reduce the number of explanatory variables, we followed variable selection procedure based on their importance [62]. In our study all variables were ranked by the decrease of %IncMSE values. We used the forward selection approach by first selecting the two most important variables to build the RF model. Then, we analyzed how the OOB error decreased by adding variables of diminishing importance at each next step. We found that the inclusion of the first 10 predictors decreased OOB error from 58% to 28% (Supplementary Figure S2a). The lowest OOB error (25%) was obtained after inclusion of 36 most important variables for the combined mosaic (Supplementary Figure S2b). About one-third of selected features was extracted from the April–October mosaic. Longitude was revealed to be the first important predictor. This also proved that the inclusion of thermal bands of any mosaic did not improve the classification accuracy of a combined mosaic. We also used a forward selection technique for each seasonal mosaic and found that the combined dataset performed the best, and was thus selected for further classification.

## 3. Results

### 3.1. Global Forest Maps

An aggregated sample of three climatic zones and groups of forest stands is shown in Figure 4. This illustrates that the majority of deviations in the canopy cover of coniferous and deciduous forest stands occurred in Polissya and steppe regions. The tree cover threshold for hardwood deciduous forests that have marginal values of relative stocking are reasonably higher, and thus it is likely that understory vegetation was captured by satellite images.

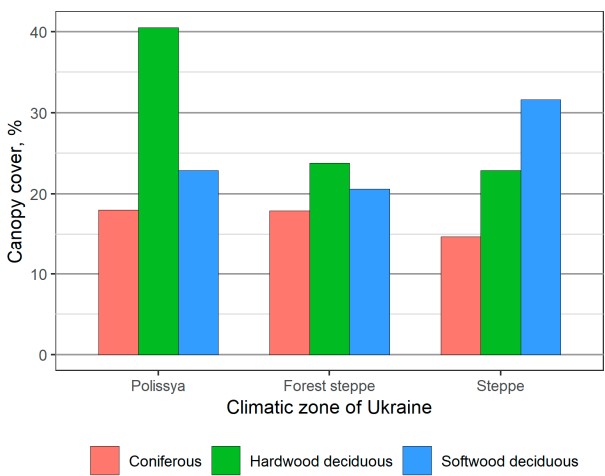

**Figure 4.** Canopy cover of forest stands with the relative stocking of 0.3–0.5.

Based on the presented data, no conclusion can be drawn about what level of tree canopy must be applied to effectively map flatland forests in Ukraine. Therefore, we tested the performance of the GFC map by applying 30% and 40% thresholds for classifying tree cover according to the distribution of forested areas by tree species composition and its proportion in different climatic zones of the study area. The results for each scenario were then compared to official data on forested area that were reported by the State Forest Resources Agency of Ukraine for 2011 [66]. Data for each administrative region (oblast) are presented in Figure 5.

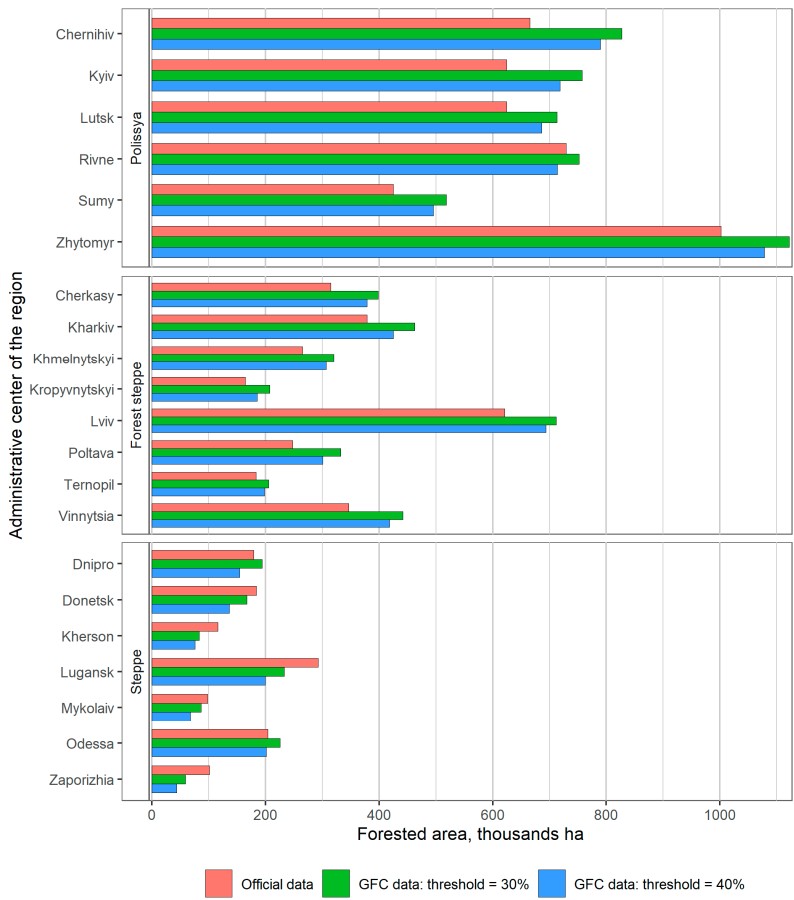

**Figure 5.** Regional estimates of forested area applying 30% and 40% thresholds for classification of GFC datasets.

Figure 5 shows that the 40% threshold applied to canopy cover of the GFC map overperforms the 30% threshold, and represents a better agreement between mapped areas and official statistics. The GFC map overestimated forested areas in almost all of the Polissya subregion while underestimating it in the steppe zone. Official data include only areas classified as forest based on the national definition. However, we understand that urban forests and forests that have regenerated naturally, for example, could substantially contribute to the forested area estimated using earth observation data.

Based on a stratified random sample of 4700 reference points, we estimated the site specific accuracy of the GFC data for each region of the study area. We found that user's and producer's accuracy have latitudinal gradients and tend to decrease from north to south. The lowest producer's accuracy (0.3–0.4) is observed for regions where forests occupy less than 5% of the territory (Zaporizhia, Mykolaiv and Dnipro oblasts). The accuracy of the GFC map for the forest steppe regions varied between 0.7–0.9. The higher values (0.8–0.95) of both producer's and user's accuracies were observed for the Polissya region.

We compared the spatial accuracy of discrete JAXA FNF and GlobeLand30 maps to forest masks extracted for GFC and LTCCF through applying corresponding canopy cover thresholds of 40% and 25%. Based on the error matrices, we performed a site-specific accuracy assessment, which includes Kappa statistics, of the overall user and producer accuracies [67] (Table 3).

**Table 3.** Accuracies of satellite-derived forest masks for flatland of Ukraine.

| Product Name | Kappa | Thematic Accuracy % | | | | |
|---|---|---|---|---|---|---|
| | | Overall | User | | Producer | |
| | | | Forest | Non-Forest | Forest | Non-Forest |
| JAXA FNF | 0.551 | 80 | 56 | 95 | 87 | 78 |
| GlobeLand30 | 0.500 | 82 | 65 | 87 | 58 | 90 |
| GFC | 0.614 | 83 | 60 | 97 | 93 | 80 |
| LTCCF | 0.667 | 86 | 68 | 95 | 86 | 87 |

We found that the most sensitive methods/RS forest product for identifying forests was the GFC map (producer's accuracy = 93%) while the least sensitive was the GlobeLand30 (producer's accuracy = 58%). Nevertheless, all forest masks included commission errors that are characterized by user's accuracies values of 56–68%. The most common confusion occurred between forested areas and grasslands with scattered trees, and un-stocked forest stands and shrublands. In this case, the JAXA FNF mask had the lowest accuracy. Examples of the performance of forest masks in different landscapes over the study area are shown in Supplementary Figure S3.

*3.2. Landsat-Based Forest Mask*

3.2.1. Forest Mask for Flatland UKRAINE

We performed the classification of seasonal mosaics on the GEE platform using the selected 36 covariates Supplementary Figure S2b. Forest cover mapping refers to the classification of spatially continuous landscapes into discrete classes. Any remote sensing-derived map can potentially contain some errors. Thus, the correction of pixel area estimates is important to derive adequate forested area. While mapping forests, we assumed that any land cover class contains some percentage of misclassified pixels associated with forests and vice versa (i.e., a forest is expected to elicit more confusion with wetlands rather than with other water bodies). In order to propagate the uncertainty of land cover estimates having different weights into forest mask [21], we used the multiclass classification rather than binary approach. As a result, we created a thematic map for each administrative region that included eight land cover classes predefined on an image interpretation phase. Finally, this was transformed into a binary forest mask of flatland Ukraine (Figure 6).

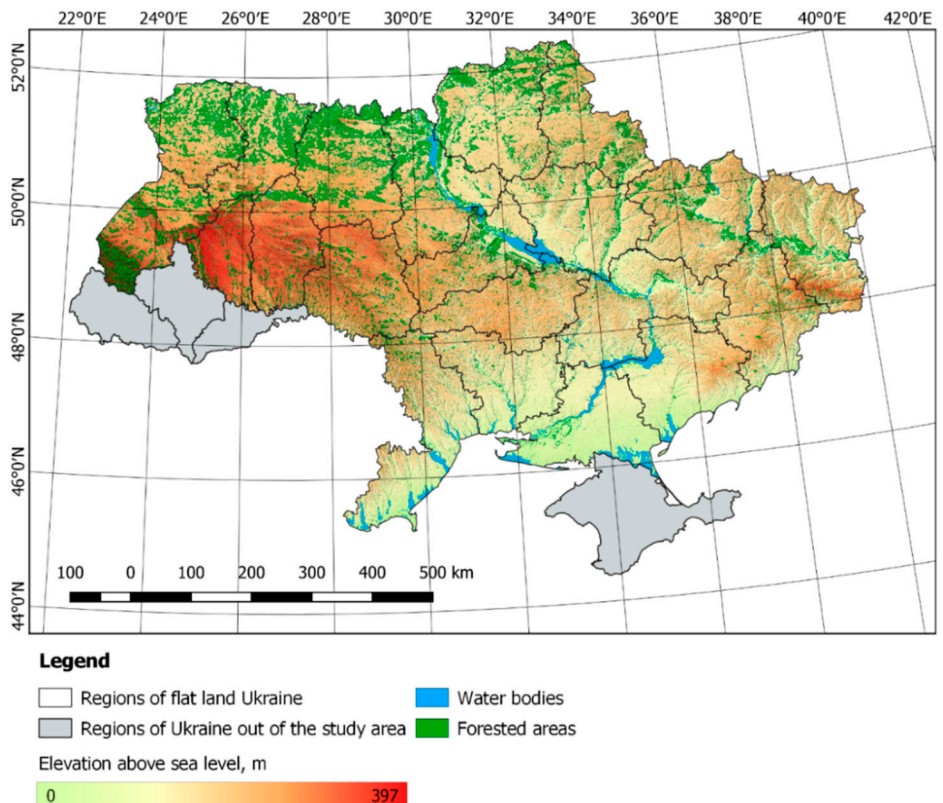

**Figure 6.** Forest mask of flatland Ukraine derived from classification of combined composited mosaics of Landsat 8 OLI images.

The forest mask indicates that flatland forests in Ukraine are distributed unevenly. The forest cover of the northern regions is significantly higher. We have also defined that most forests in steppe and forest-steppe climatic zones are associated with river valleys. This, in turn, suggests a need to focus sustainable forestry practices at the watershed scale, which can generate more ecosystem services to a greater portion of Ukrainian society.

### 3.2.2. Accuracy Assessment of the Landsat-Based Forest Mask

Accuracy assessment of Landsat 8 OLI forest mask was performed using an independent stratified reference dataset that includes 9237 points randomly distributed into eight land cover classes. Table 4 depicts the confusion between thematic classes.

We used Olofsson et al.'s [21] guidelines on "good practice" for revising the accuracy of thematic maps and the propagation of their uncertainties to area estimates. The error matrix, expressed in terms of proportion of area (Supplementary Table S1), allows us to estimate the adjusted area of class "forest" that corrects map errors. There are two forested area proportions that can be derived from the error matrix. The row total represents the proportion of mapped area ($\hat{p}_{k.}$) while the column total ($\hat{p}_{.k}$) is the real proportion of that area determined from reference classification. Conceptually, the proportion $\hat{p}_{.k}$ is a subject of sampling variability, thus has a smaller bias than the proportion $\hat{p}_{k.}$, which is likely biased due to classification errors. The map class area should be estimated using the proportion estimated from the reference classification $\hat{p}_{.k}$. We also used the "good practice" guidelines for calculating the standard error of the proportion of area to build 95% confidence intervals for forested area estimates, as well as confidence intervals of user's and producer's accuracies.

**Table 4.** The error matrix of Landsat-derived land cover classification over flatland Ukraine.

| Map Class | Reference Class | | | | | | | | Total |
|---|---|---|---|---|---|---|---|---|---|
| | **Water Bodies** | **Wetlands** | **Settlements** | **Other Unproductive Lands** | **Croplands** | **Grasslands** | **Shrubland** | **Forests** | |
| Water bodies | **859** | 23 | 3 | 13 | 0 | 0 | 1 | 1 | 900 |
| Wetlands | 34 | **744** | 15 | 2 | 14 | 17 | 24 | 50 | 900 |
| Settlements | 2 | 4 | **774** | 32 | 45 | 16 | 7 | 18 | 898 |
| Other unproductive lands | 1 | 3 | 475 | **412** | 4 | 0 | 1 | 2 | 898 |
| Croplands | 0 | 11 | 4 | 0 | **2048** | 138 | 6 | 25 | 2232 |
| Grasslands | 0 | 66 | 13 | 7 | 148 | **801** | 47 | 94 | 1176 |
| Shrublands | 1 | 62 | 37 | 0 | 161 | 101 | **341** | 197 | 900 |
| Forests | 3 | 35 | 6 | 0 | 23 | 34 | 19 | **1213** | 1333 |
| Total | 900 | 948 | 1327 | 466 | 2443 | 1107 | 446 | 1600 | 9237 |

The mapping has shown that forests occupy 17.4 ± 0.4% of Ukrainian flatlands, or 9440.5 ± 239.4 thousands ha. The Landsat-derived forest mask overestimates forested area compared to official data (14.4% and approximately 776.7 thousand ha, respectively) by about 3%. We refer this fact to different interpretations of forests during satellite images classification and the national definition of forests. Some overestimation may be the result of the MVC procedure for image composing applied for the 2014–2016 timeframe. Since this procedure preferentially selects pixels covered by vegetation, some forest loss that had occurred during this period might be treated as forest. The user's accuracy of the developed forest mask for flatland Ukraine is estimated to be 0.910 ± 0.015, while the producer's accuracy is lower: 0.880 ± 0.018. The land cover map showed an overall accuracy of 0.877 ± 0.008. However, we revealed the zonal gradient in accuracy distribution can be seen from accuracies of regional estimates of forested areas, which is presented in Table 5.

**Table 5.** Regional estimates of Landsat-derived forested area in flatland Ukraine (at 95% confidence level).

| Capitals of the Administrative Regions (Oblasts) * | Forested Area, Thousands ha | | Adjusted Proportion | User's Accuracy | Producer's Accuracy |
|---|---|---|---|---|---|
| | Mapped | Adjusted | | | |
| *Polissya climatic zone* | | | | | |
| Chernihiv | 817.9 | 803.6 ± 51.2 | 0.251 ± 0.016 | 0.947 ± 0.051 | 0.963 ± 0.036 |
| Kyiv | 707.2 | 765.4 ± 55.1 | 0.264 ± 0.019 | 0.966 ± 0.039 | 0.893 ± 0.056 |
| Lutsk | 723.2 | 717.8 ± 44.3 | 0.356 ± 0.022 | 0.936 ± 0.046 | 0.943 ± 0.038 |
| Rivne | 748.3 | 731.8 ± 51.0 | 0.365 ± 0.025 | 0.912 ± 0.052 | 0.933 ± 0.042 |
| Sumy | 556.4 | 558.8 ± 53.4 | 0.233 ± 0.022 | 0.912 ± 0.068 | 0.909 ± 0.062 |
| Zhytomyr | 1119.5 | 1151.9 ± 57.5 | 0.386 ± 0.019 | 0.972 ± 0.031 | 0.946 ± 0.038 |
| *Forest steppe climatic zone* | | | | | |
| Cherkasy | 383.7 | 430.2 ± 52.7 | 0.205 ± 0.025 | 0.940 ± 0.066 | 0.840 ± 0.091 |
| Kharkiv | 468.6 | 537.4 ± 73.7 | 0.169 ± 0.023 | 0.960 ± 0.055 | 0.838 ± 0.108 |
| Khmelnytskyi | 338.5 | 357.0 ± 32.5 | 0.173 ± 0.016 | 0.960 ± 0.055 | 0.909 ± 0.068 |
| Kropyvnytskyi | 211.6 | 241.8 ± 79.3 | 0.098 ± 0.032 | 0.780 ± 0.116 | 0.681 ± 0.214 |
| Lviv | 783.9 | 771.2 ± 73.0 | 0.353 ± 0.033 | 0.909 ± 0.070 | 0.924 ± 0.058 |
| Poltava | 319.2 | 323.6 ± 59.7 | 0.112 ± 0.021 | 0.820 ± 0.108 | 0.808 ± 0.123 |
| Ternopil | 219.8 | 217.1 ± 8.6 | 0.157 ± 0.006 | 0.980 ± 0.039 | 0.993 ± 0.003 |
| Vinnytsia | 450.2 | 469.1 ± 68.4 | 0.177 ± 0.026 | 0.880 ± 0.091 | 0.846 ± 0.100 |
| *Steppe climatic zone* | | | | | |
| Dnipro | 229.6 | 299.0 ± 76.8 | 0.093 ± 0.024 | 0.900 ± 0.084 | 0.693 ± 0.174 |
| Donetsk | 283.5 | 277.6 ± 65.5 | 0.103 ± 0.024 | 0.820 ± 0.108 | 0.835 ± 0.175 |
| Kherson | 60.4 | 65.0 ± 11.4 | 0.024 ± 0.004 | 0.840 ± 0.103 | 0.787 ± 0.119 |
| Luhansk | 334.8 | 380.6 ± 69.2 | 0.140 ± 0.025 | 0.880 ± 0.091 | 0.777 ± 0.128 |
| Mykolaiv | 90.6 | 137.0 ± 56.9 | 0.057 ± 0.024 | 0.900 ± 0.084 | 0.594 ± 0.245 |
| Odessa | 185.2 | 260.4 ± 74.9 | 0.078 ± 0.022 | 0.900 ± 0.084 | 0.640 ± 0.181 |
| Zaporizhia | 79.6 | 88.3 ± 17.7 | 0.032 ± 0.006 | 0.820 ± 0.108 | 0.747 ± 0.135 |

* Every region (oblast) has the same name as the capital city except Lutsk (Volyn oblast) and Kropyvnytsky (Kirovohrad oblast) cities.

In general, the produced forest mask significantly overestimates official forested areas for regions of Polissya climatic zone, in some instances by up to 150,000 ha. The map showed both the user's and producer's accuracies greater than 90%, indicating that there were not substantial commission and omission errors. Thus, we conclude that Landsat 8 OLI composited mosaics perform well in areas with a high proportion of forest cover. While the proportion of forest cover decreases, the producer's accuracies become lower than user's accuracy, indicating that omission errors of forested area tends to increase. In fact, we received the lowest producer's accuracy values (< 0.75) for the steppe zone, especially for regions with less than 6% forest cover (Mykolaiv, Odessa, Dnipro, Zaporizhia, Kherson). Moreover, adjusted forest area for the last two listed regions are underestimated in comparison to the official inventory data [66] by up to 50,000 ha. Besides the spatial resolution of Landsat data, which has proven to be challenging for depicting narrow windbreaks (with 10–20 m width), the underestimation

of forested area can result in the degradation of windbreak systems in some southern regions of Ukraine (i.e., steppe zone).

Based on the regional and zonal distribution of forested area estimates, we conclude that the adopted approach could be applicable for northern regions of Ukraine where forests occupy more than 20% of the area. A good example of how forest cover percentage impacts the accuracy of forest mask is Lviv oblast that we assigned to the forest steppe zone, however, it has 35.5 ± 3.3% of forest cover based on our estimates. The forest cover of the forest steppe zone varies between 10–15% and forests here are fragmented as small patches near rivers flood plains. Nevertheless, we conclude that spatial resolution of Landsat data is enough to map forests over the forest steppe zone. The forest cover in the steppe zone decreases dramatically, which results in rather more fragmentation than the other zones. As expected, we found that the lower proportion of forest cover reflects the lower producer's map accuracy.

## 4. Discussion

### 4.1. The Consistency of Tree Cover Estimates

As many studies have demonstrated, canopy cover is a primary characteristic that is important for mapping forests. Proportional per-pixel estimates of tree cover allow us to characterize forested areas in a continuous form, which better represents the vegetation canopy mosaics resulting from natural factors or human activity. Tree canopy cover maps could be transformed into discrete ones by applying an adequate threshold. As a result, the use of such datasets is of great importance for areas lacking more reliable national maps [20]. In addition to our study, the accuracy of percent tree cover maps has been demonstrated in previous studies [58,68]. However, the LTCCF is an analog of GFC and thus does not include data on forest cover changes.

There are some possible reasons why forested areas in Polissya appeared to be higher than the official data. The discrete forest mask included all woody vegetation that was not formally included in the forested area, yet interpreted from remote sensing data as forests. Following the collapse of the Soviet Union, many non-productive agricultural fields were abandoned and gradually recovered into forested areas through natural regeneration. This, coupled with expanding urban forests, was unaccounted for in the official data, thus resulting in 6–8% of the overestimations [66]. The opposite appears to occur in the southern and eastern steppe regions. We ascribed an underestimation in forest cover to the spatial resolution of the GFC map, which does not match with narrow linear windbreaks that contribute to most of the forested area in the steppe regions. This has resulted in the degradation forested areas with canopy densities decreasing over the last few decades.

We used 4700 sample points of the training dataset to assess the congruence between both products. Figure 7 shows that tree cover estimates provided by GFC and LTCCF data are in accordance, yet the variability of data is narrower for LTCCF and attains saturation when the tree cover reaches 50%. As a result, the threshold of 40% was adopted for the GFC as a criterion for mapping forested and non-forested areas should be lowered down to 25%.

A more detailed comparison of the performance of GFC and LTCCF datasets for the flatlands in Ukraine is shown in Supplementary Figure S4. According to GFC data, only a small portion of the sample size that fell within the forested areas has tree cover lower than 40% (Supplementary Figure S4a) and most observations collected on non-forested areas have zero canopy cover (Supplementary Figure S4b). We believe the threshold of 40% is a good choice for the classification of the GFC dataset since this value provides a good baseline from which the number of correctly classified data sufficiently increases. In addition, some attention should be paid to the concentration of forest-related reference data near point 0% (Supplementary Figure S4a).

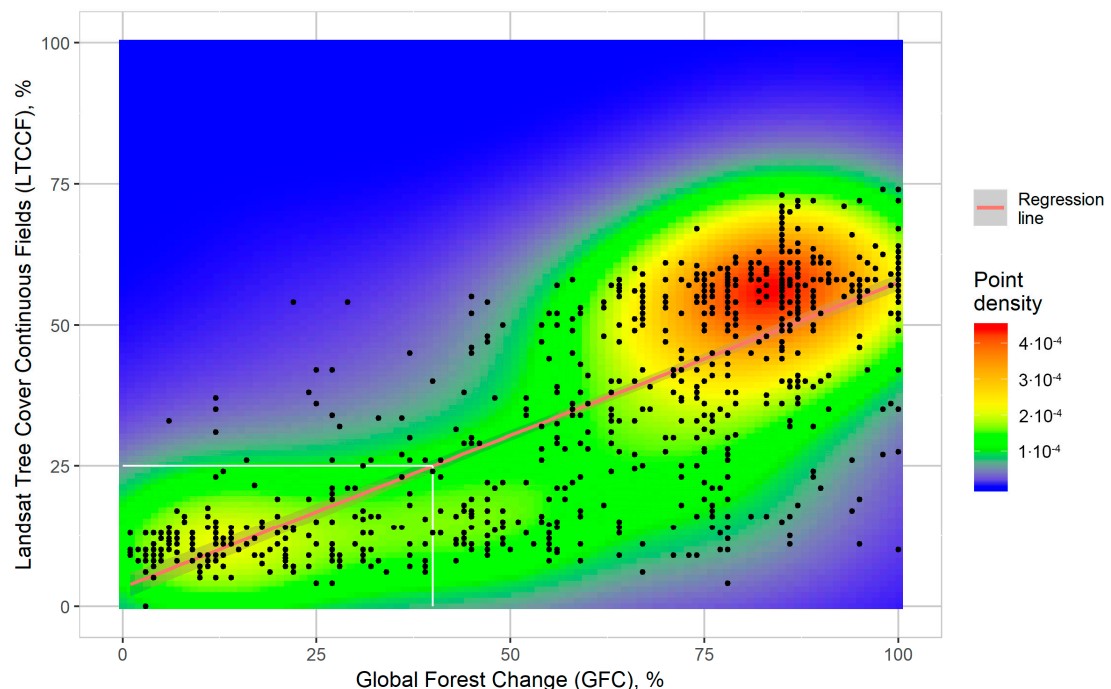

**Figure 7.** Scatter plot of percent tree cover estimates of GFC against LTCCF products.

In contrast, the quality of the LTCCF dataset is much lower. Firstly, it incorrectly represents tree cover for forested areas in a range that does not exceed 80% (Supplementary Figure S4c) and a small proportion of the sample size that is interpreted as non-forested areas has a canopy cover equal to 0% (Supplementary Figure S4b). We refer this drawback to the methodology of rescaling MODIS vegetation continuous fields to Landsat-based resolution in a severely fragmented landscape of our study area [4].

According to GFC data, about 60% of the forest-related observations having 0% tree cover are windbreaks. This means that most windbreaks on the steppe zone were not identified by GFC. The situation is more critical with LTCCF dataset in which nearly all linear shelterbelts were classified as non-forested areas. Supplementary Figure S5 represents the distribution of observation from reference dataset that were interpreted as windbreaks depending on tree cover value for both GFC and LTCCF datasets. This, in turn, provides some explanation as to why the forested area in the steppe zone of flatland Ukraine shown in Figure 5 had been underestimated by GFC data in comparison with the official state forestry data.

Thus, we concluded that in the case with GFC datasets the commission errors most likely would appear for the range of tree cover of 45–60% when non-forested areas would be classified as forests. The omission errors address mainly the misclassification of windbreaks and are typical for southern regions of flatland Ukraine. However, this drawback is more associated with the LTCCF dataset that lacks a precise estimation of tree cover for small patches of forests, i.e., <1 ha.

Our analysis shows the advantages of GFC dataset over LTCCF in that: (i) it characterizes forest canopy cover in a range of 0–100%; (ii) tree cover for most of the non-forested areas is realistically estimated to be 0%; and (iii) it is associated to lower commission and omission errors. It is also important to note that GFC maps have become popular because they represent forest change. In using these maps, we found that user's accuracy for forest loss as estimated by GFC data was 70% while for forest gain was only 42%. We also found that the forest gain layer is not precise enough if a signal combination is as follows: loss → gain.

One of the major disadvantages of the mask is that it classifies irrigated croplands on the steppe (Kherson, Zaporizhia oblasts) as forest. This problem is quite common for all products except the Globeland30 map, which was improved using an expert knowledge-based verification approach [2].

None of the global maps analyzed in the study is effective for mapping narrow (10–20 m) windbreaks. The misclassification errors in forest steppe zone mainly refer to orchards that are included in forest masks. The continuous forest maps GFC and LTCCF are more precise under fragmented landscapes and perform well in urban environments and mosaic patterns created by clear-cut harvesting. Nevertheless, their accuracies are not enough for mapping wetlands, especially those located in deltas of major rivers (e.g., Dnipro, Dnister).

### 4.2. Seasonal Dynamics of Spectral Features

The classification of different vegetation land cover is generally used as a prerequisite for extracting the correct forest mask from satellite images. Our study reveals that the time series of satellite observations track phenology changes. We combined all selected Landsat 8 OLI images by months and extracted median values for each spectral band and band combinations. Supplementary Figure S6 represents the yearly dynamics of six reference land cover classes in band combinations of red and infrared spectra as an example. Most of these band combinations were associated with unique spectral profiles. For example, the ratio of red (Band 4) and near-infrared (Band 5) spectral ranges of Landsat images were revealed to have the lowest value for forested areas during a yearly timeframe. In the classification process, "forest" was also distinguished well in Band 6/Band 7 ratio as having higher values for nearly all months in a year except January. This may be due to snow cover on croplands that have the highest values. After plotting spectral profiles of selected land cover classes for all spectral features used in the study (Table 1, Supplementary Figure S2b), we conclude that most of them contribute to the total accuracy of classification. We also found that thermal infra-red data (Band 10) is essential for land cover classification because of differences in temperature of water bodies, infrastructure, forested and non-forested areas.

We analyzed the potential for discrimination of coniferous, deciduous, and mixed forest stands using time series of Landsat 8 OLI satellite images. The reflectance of these forest groups also has a specific trend over the year. Coniferous and deciduous forests could be identified with high accuracy during leaf-off periods in winter, early spring (March, April) and late autumn (October, November). The rather opposite trends in the spectral reflectance of coniferous and deciduous stands are captured in Band 4 and Band 5 of Landsat time series. Although the discrimination of tree species in the red band is insufficient, we noticed that this improved when deciduous forests have no green biomass and refer to higher values in Band 4 (Supplementary Figure S6c). The reflectance of coniferous tree species in the near-infrared spectrum (Band 5) dominates during the leaf-off period, albeit with the start of vegetation phase reflectance of deciduous species that increases more rapidly (Supplementary Figure S6d). Spectral reflectance of coniferous stands is higher than deciduous ones over the year in both short-wave infrared Landsat bands (Bands 6 and 7). We found that thermal bands for identification of abovementioned groups of tree species did not significantly reduce the classifier error. Mixed forest stands as they were identified in this study tend to have intermediate reflectance values but likely to produce confusion with both coniferous and deciduous stands. This problem requires the application of more advanced methods for predicting tree species composition using sample-based forest inventory data [69,70].

Classification accuracy could not be assessed directly from Table 4 because the sample counts from different map classes have different weights. Based on our use of Olofsson et al.'s [21] guidelines, we found that sample counts that populate the error matrix should be substituted by area proportions that correspond with them in the forest cover map. The known area and proportion of each stratum should be accounted for to describe sampling design. The forest mask provides different accuracies for Polissya, forest steppe, and steppe climatic zones of Ukraine. Thus, in the north of Ukraine (Polissya), both producer's and user's accuracies ranged from 0.90 to 0.97 and tend to decrease further south (forest steppe and steppe zones). In the case of Landsat-based forest masks, we found the same tendency in underestimation of forested areas in the steppe zone (Zaporizhia, and Kherson oblasts). It was revealed again that the lowest accuracy (0.6–0.75) is associated with regions where forests occupy

less than 6%. As a result, we concluded that the Landsat-based mask has higher accuracy compared to those extracted from global datasets but performs well only in Polissya and Forest-steppe climatic zones of Ukraine. For these zones, the forest mask captures fragmented forest patterns.

Our study shows that on the regional scale, due to fragmented and unevenly distributed forests, global products provide forest cover masks with various levels of uncertainty. The continuous forest maps GFC and LTCCF are recommended for the rough estimation of forest cover in the flatlands of Ukraine rather than discrete products such as GlobeLand30 and JAXA FNF. However, it is critically important to choose the right canopy cover threshold value for identifying forest stands clearly. Similar studies have found that GFC and LTCCF do not perform well for the accurate description of regrowth that usually follows disturbances. Spatial patterns of forests and inconsistent forest inventory statistics at the state level may cause a significant underestimation (up to 30% less compared to official data) of the forested area [71]. Thus, the effective mapping of forest successions plays an important role in assessing forest characteristics in severely fragmented forest cover, such as those in flatland Ukraine. We also acknowledge that the Landsat-based forest mask we developed for this research has similar limitations. Therefore, we suggest higher spatial resolution satellite images coupled with advanced approaches for image time series processing [12,72] should be employed to increase the accuracy of forest mapping in the steppe zone. In addition, considering topographic information from digital elevation models could improve forest type mapping as it provides axillary data in terms of the spatial pattern of forest distribution.

## 5. Conclusions

This study addresses the specific issues of forest mapping under fragmented landscapes in the flatlands of Ukraine. We assessed the site-specific accuracy of a global forest mask having 25–30 m spatial resolution and compared it to Landsat-based classification. Based on the derived results, several conclusions could be drawn. First, global forest masks lack accuracy for the precise quantification of severely mosaiced forested areas. Continuous forest cover products GFC and LTCCF overperform the discrete analogues in terms of accuracy because provide options for selecting optimal tree cover threshold value. Second, Landsat 8 OLI images are the primary source of spectral data used for forest classification because they allow to realize sophisticated algorithms for image classification. Phenology-based methods have revealed to be robust for estimation of forested areas for our study and provided an accuracy of about 88–91%, which we believe is satisfactory at the regional scale. We found that Landsat-derived forest mask performs with high levels of both user's and producer's accuracies for Polissya and forest steppe climatic zones of Ukraine where forests occupy more than 15%. However, we encourage the employment of satellite images of higher spatial resolution for areas that have a low level of forest cover. Thus, the GEE platform enhances the image processing routine, allowing for the performance of more complex procedures of data composition, classification, and accuracy assessment at a large scale.

**Supplementary Materials:** The following are available online at http://www.mdpi.com/2072-4292/12/1/187/s1, Figure S1: Examples of collection of reference data using Google image analysis in time: (a) the plot is interpreted as unforested (Lon = 33,370,047; Lat = 51,672,211); (b) the plot is classified as forested (Lon = 29,324,394; Lat = 51,324,563); (c) for the period of 2010 the plot is forested but (d) in 2015 is classified as unforested (Lon = 29,060,559; Lat = 49,968,467); (e) in 2004 the plot is unforested whereas (f) in 2015 is interpreted as forested (Lon = 34,403,828; Lat = 50,597,736). Figure S2: Impact of predictor variables onto the classification accuracy (panel a) shows the relationship between the number of variables and out-of-bag classification error, (panel b) shows the ranked predictors' impact onto the decrease of MSE of a model). Figure S3: Examples of global forest masks performance over different landscapes of flatland of Ukraine: (a) irrigated croplands (Kherson oblast, Lon = 33.8736, Lat = 46.5905); (b) wetlands (Odessa oblast, Lon = 30.0580, Lat = 46.4889); (c) water bodies (Volyn (Lutsk) oblast, Lon = 24.1981, Lat = 51.4958); settlements (Kyiv oblast, Lon = 30.6755, Lat = 50.4186); (e) clearcuts, meadows, unstocked forest crops (Zhytomyr oblast, Lon = 29.5300, Lat = 50.7236). Figure S4: Distribution of reference sample units on tree cover values: (a) GFC forested areas; (b) GFC non-forested areas; (c) LTCCF forested areas; (d) LTCCF non-forested areas. F represents samples that would be classified as forests, NF—non-forests if a threshold of 40 % is applied to GFC and 25% to LTCCF datasets. Figure S5: Distribution of tree cover values of windbreaks: (a) GFC; (b) LTCCF. F represents samples that would be classified as forests, NF—non-forests if a threshold of 40 % is

applied to GFC and 25% to LTCCF datasets. Figure S6: Examples of monthly dynamics of spectral features of Landsat 8 OLI time series: (a,b) land cover classes; (c,d) groups of forest stands. Table S1: The error matrix of Landsat-derived land cover classification over flatland Ukraine expressed in terms of the proportion of area as suggested in Olofsson et al. (2014).

**Author Contributions:** Conceptualization, V.M. and A.B.; methodology, V.M. and M.K.; software, V.M.; formal analysis, V.M., A.B., S.L., A.J.S. and M.K.; resources, V.M., M.K.; data curation, V.M., M.K. and A.B.; writing—original draft preparation, V.M., M.K. and A.J.S.; writing—review and editing, V.M., A.J.S., S.L. and M.K.; visualization, V.M. All authors have read and agreed to the published version of the manuscript.

**Funding:** This research received no external funding.

**Acknowledgments:** We would like to thank the National Geomatics Center of China (NGCC) for providing us with Globeland30 products for our study. We would also thank the Ukrainian Government Project Association (PA "Ukrderzhlisproekt") for providing us with national forest inventory database. The authors also would like to acknowledge Dmytro Gilitukha for his contribution to analysis of GFC map.

**Conflicts of Interest:** The authors declare no conflict of interest.

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
