# Peer review of "Regional-Scale Forest Mapping over Fragmented Landscapes Using Global Forest Products and Landsat Time Series Classification"

_remotesensing, doi:10.3390/rs12010187_

Round 1

Reviewer 1 Report

I wanna thank the authors for kindly accepted the suggestions.
This study is a great contribution for the forest mapping, nevertheless I still have some questions/suggestions:

1) Technical Approaches:

1.1) In the lines 314-317 you talk about the reference dataset used in the accuracy assessment of the Landsat-based forest mask (i.e. 460 x 8 = ~3680 sample points). In the line 438 you talk about the dataset used in the accuracy assessment of global forest maps (i.e. 4700 reference points). In the line 482, and in the Table 4, you used 9237 random points as reference dataset. I really got confused by all these numbers that don't talk to each other. After all, what is the size of your reference dateset ?

1.2) You claim the Landsat-based forest mask got a higher accuracy compared to the global forest products (line 677), but you didn't use the same reference dataset to evaluate all these maps, right ?. To conduct an fair comparison between them you need to use exactly the same reference dataset of forest samples (e.g. 1.600 forest points of the table 4), and preferably put a new row in the table 3 explicitly pointing the accuracy of Landsat-based forest mask.

2) Manuscript Structure:

2.1) The lines 398-416 should be moved to section "2. Materials and Methods"

Author Response

Please, find our responses in the attached file below.

Reviewer 2 Report

Stripes 247-253: Justify why it was decided to use these bands or combinations of bands for the characterization of phenology and classification.

Stripes 291-294: The statistics used for the sizing of samples and the calculation of accuracy should be better explained

In table 5 the regions should be inserted to better read the data

A clear definition of the dimensional characteristics of the windbreaks is lacking, and of how and if the Landsat images, the GFCs and the LTCCFs are able to map them. The dimensions of these elements would not allow their mapping using satellite images above 15-20 meters of resolution. The official data of the forests of Ukraine by the NFI should not consider these elements because they define as wood the wooded areas wider than 20 meters. So I don't know if it is possible to consider these surfaces in the net balance of the surfaces mapped by forest masks.

Author Response

(The authors gave the same response as above.)

Round 2

Reviewer 1 Report

Thank you for kindly accept the suggestions. This study is a great contribution to the forest mapping and can be accepted in the present form.

This manuscript is a resubmission of an earlier submission. The following is a list of the peer review reports and author responses from that submission.

Round 1

Reviewer 1 Report

First of all, congratulations for the work, the text flows-well, the introduction is great, the supplementary material complements the manuscript and this type of regional-scale analysis is really relevant.

Here is my suggestions:

1) General Aspects:
1.1. The biggest issue of the manuscript is produce a multi-classes map to extract only the forest class. Conceptually, it is a really strange and misleading solution, because a production of a multi-classes maps is more challenging than a production of binary maps. You need justify well this methodological decision in the text. Otherwise, You should considering produce a forest binary map or even a probability map through Google Earth Engine (see PARENTE & FERREIRA, 2018)

2) Manuscript Structure:
2.1. The abstract could be better. You should considering include some parts of introduction to justify the relevance of the work.
2.2. Considering include a workflow figure, in the section 2, with all your data and implemented approaches (e.g. forest cover threshold analysis, Random Forest sensitive analysis). It will help the readers to understand all you did.
2.3 A very practical contribution of your work was the elaboration of a protocol to establish a forest cover threshold, considering these global products, at regional-scale. Considering include a specific section about it, clearly indication how other researcher could apply it in other parts of the world.

3) Technical Approaches:
3.1. Why you used a cloud scoring algorithm rather than Landsat QA-Band ? This decision should be well justified in the text. (line 330)
3.2. It's confusing compare the OOB errors from different seasonal mosaics considering different values of ntree parameter. For example, maybe the combined seasonal mosaic produced a best result because of larger number of trees and not for the feature space itself. You should use the same ntree for conduct this sensitivity analysis (My suggestion is 100 ntree - see BELGIU & DRĂGUŢ, 2016). (line 330)
3.3. How you defined the values for mtry ? The square root of the feature space size ? (line 330)
3.4 What are the 36 most important variables for the combined mosaic ? Why you choose 36 ? (line 336)
3.5 You conducted the accuracy assessment considering ~4730 samples presented in the table 5. Does samples are the same that was used to training Random Forest ? If yes, all your accuracy assessment is wrong and overestimated, because you shouldn't use the same samples to train, validate and test your model. In this scenario, you are doing a very unfair comparison between your classification result and other forest cover products. To do it right, you should considering completely independently reference data in the both comparisons.
3.6. How you estimated the user’s accuracy for forest loss and forest gain using only one observation date of Collect Earth ? Even looking the same year of the forest loss event (e.g. 2015), nothing guarantees that the image analysed by you was acquired in the same time-frame that Landsat image analysed by GFC, right ? How you handled that ?

References:
- BELGIU, Mariana; DRĂGUŢ, Lucian. Random forest in remote sensing: A review of applications and future directions. ISPRS Journal of Photogrammetry and Remote Sensing, v. 114, p. 24-31, 2016.
- PARENTE, Leandro; FERREIRA, Laerte. Assessing the spatial and occupation dynamics of the Brazilian pasturelands based on the automated classification of MODIS images from 2000 to 2016. Remote Sensing, v. 10, n. 4, p. 606, 2018.

Reviewer 2 Report

General feedback

The manuscript entitled „Regional-scale forest mapping over fragmented landscapes using global forest products and Landsat time series classification” addresses a relevant question on mapping of forested areas over fragmented landscapes in Ukrainian flatland using Landsat data, as well as regional-based accuracy assessment of some existing global forest cover products. The authors collected a set of reference data, which were used for 1) calibrating an RF classifier to distinguish between eight land cover classes and 2) an regional-scale accuracy assessment of four global forest cover products (Global Forest Change data, Landsat Tree Cover Continuous Fields, Global PALSAR-2/PALSAR Forest/Non-Forest map from Japan Aerospace Exploration Agency, and GlobeLand30 product).

The research topic is relevance and is in the scope of the journal. However, the methods need to be reconsidered and the manuscript needs to be substantially improved and restructured.

Major comments:

In my opinion, it is too ambitious from the authors to draw some critical conclusions regarding the under- or overestimation of forest cover fraction by Ukrainian official reporting when comparing the forest cover fraction derived from the Global Forest Change dataset with this from the Forest Resources Agency of Ukraine. When doing such comparisons, the authors need to take into account at least this two aspects: 1) what is defined as “forest” in Ukraine (forested areas with a “relative density” 0.3) and what was actually mapped in the 2000 Percent Tree Cover from the Global Forest Change dataset (vegetation taller than 5 m). The Ukrainian definition include no information on the minimal tree/shrub height. On the other hand, the 2000 Percent Tree Cover includes trees outside forest (urban areas); 2) time lag between these two datasets.

The soundness of the accuracy assessment need to be clarified. What data was used as reference? The 4731 sample plots with a size of 0.5 ha (50 x 50 m?) collected by visual interpretation of satellite images from Google Earth tool? From which year are these images? What is the time lag between reference data and the corresponding global forest cover products? What was the criteria to stratify the plots in “forest” and “non-forest” (see Table 1)? How the sample plots were assigned to the forest cover masks? How did the authors deal with differences in spatial resolutions of the global forest cover products (for example, Global Forest Change dataset has a spatial resolution of 30 m, whereas the Global PALSAR-2/PALSAR Forest/Non-Forest map has a resolution of 25m)? In addition, it needs to be taken into account that such statistics as overall accuracy are highly sensitive to class imbalance.

An own land cover map for Ukraine was produced using Random Forest classifier trained on variables derived from Landsat 8 data. The authors trained the RF on different variable sets: 1) spectral data, 2) spectral and geographic coordinates (Table 3). However, including geographic coordinates in such classification tasks is an absolute no-go. It is not also clear, how the classification models were validated. Where the data in Table 4 come from? From a cross-validation?

The manuscript need to be strongly improved. In the introduction, a wider review on available forest cover products (global, as well as regional, e.g. products for other countries), used sensors (Sentinel data, aerial images, Lidar) as well as algorithms is needed. The four global forest cover products evaluated in the study need to be described in a separate section in details. The approach for accuracy assessment need to be described in the Methods section. Parts in Results need to be reconsidered to Methods (LL. 259-275, Sections 3.2.1, 3.2.2., LL.323-329)..

Reviewer 3 Report

The manuscript describes a classification method for mapping forest in Ukraine. The remote sensing of land cover land use change is great importance, but in my opinion,  the experiment of the study was poorly designed and this manuscript is not well organized. Especially, the description of methodology did not provide enough information for the readers to fully understand the method used.

1) In the abstract and introduction, the authors stated that they would take advantage of the dense time series of Landsat imageries to improve the forest mapping. However, in the following text, the benefits with dense time series of remote sensing was not demonstrated.

2) A reference data based on Landsat imageries  was generated using RF. However, only ~50 trees were used, far less than the RF recommended value ~500.  The high  OOB error may be caused by the low number of trees.  TOA reflectance and band ratios were used as features. However, the TOA reflectance is heavily influenced by the atmosphere.  Why did not the authors use surface reflectance data or NBAR data?  What are the sources of the training data for the RF, seems the training data should include the all the classes listed in Table. 4.

3) The authors firstly compared the four global forest mapping products. However, the authors failed to provide details about these four data products, e.g. spatial/temporal resolutions,   how they were  generated, are they absolutely independent with the reference data?